# Application of Physical Methods for the Detection of a Thermally Degraded Recycled Material in Plastic Parts Made of Polypropylene Copolymer

**DOI:** 10.3390/ma14030552

**Published:** 2021-01-24

**Authors:** Luboš Běhálek, Jozef Dobránsky, Martin Pollák, Martin Borůvka, Pavel Brdlík

**Affiliations:** 1Faculty of Mechanical Engineering, Technical University of Liberec, Studentská 2, 461 17 Liberec, Czech Republic; lubos.behalek@tul.cz (L.B.); martin.boruvka@tul.cz (M.B.); pavel.brdlik@tul.cz (P.B.); 2Faculty of Manufacturing Technologies with a Seat in Presov, Technical University of Kosice, Bayerova 1, 080 01 Presov, Slovakia; martin.pollak@tuke.sk

**Keywords:** recycled material, tensile properties, bending properties, rheological properties, differential scanning calorimetry, induction oxidation time

## Abstract

The paper deals with the possibility of applying physical methods to detect a thermally degraded recycled material in plastic parts made of polypropylene. Standard methods of evaluating the mechanical properties of the material under static tensile and bending stress, as well as under dynamic impact stress using the Charpy method, were used for the experimental measurements. The rheological properties of materials were monitored using a method involving measuring the melt flow index, while their thermal properties and oxidative stability were monitored using differential scanning calorimetry. Based on the methods used, it can be clearly stated that the most suitable technique for detecting thermally degraded recycled material in polypropylene is the method involving establishing the melt flow index. The bending test seems to be the most suitable method for detecting recycled material by measuring the material’s mechanical properties. Similarly to the melt volume flow rate (MVR) method, it was possible to unambiguously detect the presence of even a small amount of recycled material in the whole from measuring the material’s bending properties. It is clear from the results that in the short term, there may be no change in the useful properties of the parts, but in the long term the presence of degraded recycled material will have adverse consequences on their lifespan.

## 1. Introduction

The possibility of studying recycled materials in polypropylene through determining their thermal properties using the DSC method is assessed based on monitoring of the phase transformations of a recycled material during cooling of the melt and melting of the material. As part of the experimental measurements evaluation, the change in enthalpy of crystallization (ΔHc) and the change in enthalpy of melting (ΔHm) were evaluated, the values for which are directly proportional to the share of crystalline in a material’s structure. Furthermore, the crystallization peak temperature (Tp, c) and the melting peak temperature (Tp, m) were evaluated, the values of which, according to theoretical assumptions, are most dependent on the molecular weight of the polymer.

The paper by Al Mahmood et al. dealt with the problem of recycling polymer laminated aluminum packaging (PLAP) materials into their original metal form, which are used mainly in the production of high-strength flexible packaging. The authors dealt with the influences of several polymer laminates on the quality of Al surfaces recycled using thermal disengagement technology (TDT). Three types of flexible PLAP materials were tested at different temperatures and times. The total polymer degradation and dissociation of volatiles in the PLAP material was achieved at 550 °C within 20 min of the heating process. The result of the investigation was a finding that the oxidation rate of unlaminated aluminum is 70–90% higher compared to aluminum recycled from the PLAP material [1].

Another paper studied the possibility of using waste materials. Ajorloo et al. focused on recycled polypropylene (PP) as a material suitable for the production of automotive components. They found that increasing the share of recycled PP in the composite leads to a reduction in mechanical properties, in particular the impact strength and ductility of the newly formed material. The fabricated material also showed other significant advantages associated with increased tensile strength, and as the percentage of the recycled PP increased, the melting point of the material decreased. Based on the results of the study it can be argued that the use of percentage ratios of 20:80 to 40:60 of the recycled PP to the virgin PP in composites meets the environmental and economic impact requirements for the production of components for the automotive industry [2].

Another paper by Asensio et al. dealt with processing of pre-impregnated thermoplastic towpreg reinforced by continuous glass fibers and polyethylene terephthalate (PET) recycled using pultrusion. Globally, most waste is generated by the packaging elements of consumer goods made from PET materials. The study focuses on the use of this recycled material as a feedstock for the pultrusion process. The authors compared the processability of the final composite yields of parts of three different pre-impregnated recycled materials. These materials differed in their viscosity and flow. The research results were then compared with another pultruded thermoplastic (polypropylene) composite material [3].

The authors of yet another paper, Urtekin et al., investigated the effects of the involvement of eggshell (ES) as a filler material on the properties of softened PLA and an intumescent flame retardant (IFR) containing softened PLA. The authors ran tests using DSC (Differential Scanning Calorimetry), TGA (Thermogravimetric analysis), LOI (Limiting oxygen index), UL-94 (Classification and Flame-Retardant Thermoplastics), SEM (Scanning electron microscopy), and a tensile test. The results of the research were proof that ES can be used as a low-cost filling material in IF and PLA systems [4].

Roik et al. dealt analyzed the formation structure, mechanical and physical properties, and antifriction properties of the newly formed composite material based on the crushing of high-alloy waste from an AD35 aluminum alloy. Such material is suitable for use in the contact joints of folding machines or printing machines. The results of the study proved that the technology used for this material processing procedure is suitable for processing recycled waste by grinding AD35 aluminum alloy, and that it enables new antifriction composite materials of homogeneous structure to be obtained without flow phenomena and with the required technological parameters [5].

In their paper, Zhao et al. dealt with the intelligent injection of plastics into molds for mass production of plastic components. An important task in correct component production is optimization of the injection molding process in order to produce geometrically precise parts with specified mechanical properties. The authors pointed out that it is important to pay attention to methods and strategies that can be used for sensing, optimizing, and controlling the intelligent injection molding through feedback and machine learning, which is addressed by a number of world-renowned authors and research teams [6].

In their, the authors Orji et al. investigated the effects of the use of rice husk particles on the properties of recycled polyolefin composite materials, with the main focus on recycled high density polyethylene (rHDPE) and polypropylene (rPP), together with the addition of malleated polymer coupling agents. The composites (made by mixing natural fibers with rice husks with particle sizes of <1 mm and 0.5 mm and with recycled polyolefins, rHDPE, and rPP) were subsequently extruded using a twin-screw extruder. The mechanical, physical, and thermal properties, as well as dynamic melt flow values, were monitored. Rheological tests were also done. As a result, the presence of rice husks with a particle size <0.5 mm had a positive effect on improving the viscosity and flexural strength, resulting in better water absorption and higher thermal stability for both of the rHDPE polyolefins, while the properties proved to be more reliable compared to those of the rPP composites [7].

Today, fiber-reinforced thermoplastic composites (CFRTPC) are increasingly used as alternative materials, which will one day replace conventional materials such as thermosetting polymers and metals, due to their excellent mechanical properties. Chacón et al. assessed the ability of exactly these composites to meet the need for additive production of FDM equipment, with a focus on the influences of the print orientation, material application, and wall thickness of the manufactured parts, as well as the volumes and resulting mechanical properties of the printed components. Several were been conducted, such as tensile tests and the three-point bending test. The results of the study pointed to the advantages of composites with increased contents of carbon fibers in terms of the increased strength and stiffness of products produced using additive technology [8].

In their paper, Guo et. al. dealt with the damage, fracture, and damage rates of highly filled WF–Re-HDPE composites with different proportions of WF wood fibers (50%, 60%, and 70%). They focused on three phases, which were the onset of damage, the accumulation of damage, and destructive damage resulting in breakage. The three-point bending test technique and combined acoustic emission (AE) and scanning electron microscope (SEM) analysis were used [9].

In their paper, Kim et al. studied additive technologies using metal compounds. In their paper, the authors focused on the development of a new approach to production in the form of porous structures of bodies produced using a material created by mixing slag and reinforced metals. This material had excellent mechanical properties. The authors compared the physical properties of materials formed by mixing the slag and ceramic beads with materials created by mixing iron slag, alumina, and zirconia. Tests for the average particle size, maximum compressive stress, and bending stress were conducted, as were several stress tests. The authors also focused on lightening of the material intended for additive production while maintaining the high strength of the manufactured parts [10].

The paper by Sun et al. discussed the possibility of using waste material created using additive technology for recycling and subsequent product manufacturing via additive production. The authors discussed the ecological and economic impacts of the use of recycling in additive production. Thorough studies and the acquisition of practical knowledge have shown that it is possible to produce high-quality products from recycled materials, while at the same time the supply of such materials on the market has been evaluated [11].

## 2. Experiment Preparation

A polypropylene with the trade name SABIC^®^ 926 01 (Sereď, Slovak Republic) PP CX02-82 was selected for experimental research into the possibility of studying the detection of recycled materials in polymer ejections (plastic components). It is an emission-optimized, highly crystalline propylene–ethylene copolymer that offers high rigidity in perfect equilibrium with high temperature. The dimensional stability, impact toughness, and flow of the basic properties of the above polypropylene copolymer are given in Table 1.

### Preparation of Thermally Degraded Recycled Materials

For experimental measurements aimed at detecting recycled materials in polypropylene, conditions were simulated in which thermally degraded polypropylene was inappropriately added to virgin material due to poorly set technological parameters for injection molding. To determine the properties of SABIC^®^ PP CX02-82 polypropylene, test specimens should be made in accordance with the international EN ISO 1873-2 standard at a melt temperature of 200 °C (for the melt flow index value, see Table 1), so as to avoid undesirable cleavage of macromolecule chains due to the material’s thermal degradation [12].

To prepare the degraded material, the melt temperature was gradually increased up to 330 °C. The material’s thermal degradation was verified in the subsequent measurement of the melt flow index from the prepared crumb ejections made at different melt temperatures. The preparation of degraded polypropylene material was based on the assumption that the cleavage of macromolecule chains clearly occurs in cases where the measured value of the melt flow index increases by more than 1.5 times compared to its original value (see EN ISO 1873-2). The measurement of the melt flow index was done in accordance with EN ISO 1133-1 by establishing the so-called volume melt flow index (MVR) [13]. It is a well-known fact that chain cleavage due to polymer degradation results in its reduced molecular weight and increased flowability. The effect of melt temperature on the MVR is recorded in Table 2.

Based on the measured values, a melt temperature of 310 °C was used for the preparation of thermally degraded recycled materials as part of the experimental research. The technological conditions for injection of samples for the preparation of recycled materials are given in Table 3. The prepared samples were then crushed in a knife mill.

## 3. Preparation of Test Specimens

Prior to the experimental measurement, it was necessary to prepare the samples, which consisted of mixing the thermally degraded recycled material in the form of crumb and virgin granulates. The mixing of the components was done manually in the specified amounts, corresponding to the following shares of the recycled material in the virgin material, namely 0%, 10%, 20%, 30%, 50%, and 100%. From the mixture of materials prepared in this way, multi-purpose specimens of type A or A1 were injected according to EN ISO 3167 or according to EN ISO 527-2, respectively. The production of multi-purpose test specimens in the form of double-sided blades was carried out in accordance with EN ISO 294-1 under the conditions specified in the international EN ISO 1873-2 standard. The injection conditions are given in Table 4. The actual injection process was done on a standard Arburg 270 S 400-100 column injection molding machine with a universal two-plate injection mold. At the end of the production cycle, all test specimens were conditioned under standard conditions of 23 °C / 50 % (temperature/relative humidity) for a period of at least 40 h according to EN ISO 1873-2. Half of the samples were exposed to an elevated temperature of 150 °C for 500 h in an oven with forced air circulation before the study of their useful properties in order to simulate the evaluation of the ejection’s life depending on the recycled content.

## 4. Evaluation of Results and Discussion

This chapter discusses the possibility of verifying the presence of the recycled material in the polymer by subjecting the samples to the method of differential scanning calorimetry (DSC), i.e., by determining the thermal properties of the material and its oxidative stability, and further by determining the melt flow index (MVR) and by analyzing the mechanical properties of ejections. The influence of the presence of the recycled material on the lives of the samples was also evaluated, whereby the lives were simulated by exposing the materials to an elevated temperature of 150 °C for 500 h according to the internal regulations of the VW Group.

### 4.1. Evaluation of Thermal Properties Using the DSC Method

The possibility of studying the presence of recycled materials in polypropylene through determining their thermal properties using the DSC method was assessed based on monitoring the phase transformations of the material during cooling of the melt and melting of the material. As part of the experimental measurements evaluation, the change in enthalpy of crystallization (ΔHc) and the change in enthalpy of melting (ΔHm) were evaluated, the values of which are directly proportional to the share of crystalline in the material’s structure. Furthermore, the crystallization peak temperature (Tp, c) and the melting peak temperature (Tp, m) were evaluated, the values of which, according to theoretical assumptions, are mostly dependent on the molecular weight of the polymer.

The individual graphic representations in Figure 1, Figure 2, Figure 3, Figure 4, Figure 5, Figure 6 and Figure 7 describe the changes in enthalpy values and temperature peaks in the second phase of heating and cooling of the examined samples, depending on the weight ratio of the recycled material. Since the virgin material is a copolymer, two components corresponding to the ethylene and propylene units in the macromolecule chain appear on the thermograms (after exposure, the first component is more or less suppressed or shows random values).

From the measured values of the changes in enthalpy of crystallization and melting of the first component in the sample formed by ethylene units before exposure to elevated temperature, it can be seen that the change in the recycled weight ratio had almost no effect on changes in melting and crystallization enthalpy (see Figure 1 and Figure 2). Slight nuances can be observed within the range of statistical errors of measurement. This behavior was in agreement with data observed by Ng et al. [14].

Due to the thermal properties of the second component measured in the samples (consisting of propylene units) before their exposure (see Figure 3 and Figure 4), it can be stated that for changes in the enthalpy of crystallization and melting, these values decrease with a recycled content of 50 wt.% or more. The results also confirm the conclusions made by Capitain et al. [15].

In the case of a decrease in the values of the respective changes in melting enthalpy, it can be stated that ejections with a greater share of the recycled material weight show lower crystallinity. Differences in melting and crystallization enthalpies between the virgin material and the material containing 50 wt.% of the recycled content, however, did not exceed 5%. In terms of practical experience and with regard to methodological factors influencing the method of determining enthalpy values, it is, thus, debatable as to the extent to which it could in fact be used in the detection of a recycled material in a polymer. A difference of up to 5% can rightly be considered below the statistical significance threshold. This behavior was in agreement with data observed by Drzezdzon et al. [16].

Additionally, the melting point values and the crystallization of the first and second components in the samples before exposure to elevated temperature were identical, while the variance of the measured values was below 2%.

On the contrary, for samples exposed to an elevated temperature of 150 °C for 500 h, it can be stated that with the increased addition of recycled material, the crystallization temperature of the material decreased, even at the recycled material’s content of 10 wt.%, when a decrease of the crystallization temperature by 8% was recorded in the second and first components in the sample (see Figure 5 and Figure 6).

However, regarding the solid crystalline component in the sample, consisting of ethylene units, the measured values of the crystallization temperature were unstable, probably due to the possible inhomogeneity of the material’s degradation (see Figure 5—data are missing). Regarding the second crystalline component formed by the propylene units, it can be stated that with increased addition of the recycled material above 10 wt.%, the value of the crystallization temperature of the material did not change anymore (see Figure 6).

Similar behavior was observed when evaluating the change in enthalpy of crystallization, but the change was in the range of statistical error of measurement.

The same can also be said for the exposed samples when evaluating the melting temperature and the change in melting enthalpy. As the proportion of the recycled material increases, the melting point of the exposed samples decreases, and in the case of the second crystalline component the difference at 50 wt.% of the recycled material was ca. 6.5% (see Figure 7). The results for the thermal properties of the samples after exposure (especially for the melting points of the crystalline regions formed by the propylene units) indicate a reduced service life of the sample with an increased proportion of the recycled content contained in the virgin material.

Drawing on theoretical knowledge, it was assumed that the degraded recycled material or the cleavage of macromolecular chains (decrease in molecular weight) could be identified through a decrease in melting point (Tp, m). From the results of experimental research, it can be stated that there were no changes in melting temperature, with the exception of samples exposed to elevated temperature, where the melting point decreased by a maximum of 8%, clearly indicating a reduction in the life of an ejection with the increasing proportion of the recycled content. The results also confirm the conclusions obtained in the studies by Ewans et al. [17] and Schick [18].

### 4.2. Evaluation of Oxidation Induction Time (OIT)

From the obtained OIT values, a graphical illustration of the dependence of the oxidation induction time on the proportion of weight of the recycled material contained in the polymeric material was compiled (see Figure 8).

From the course of OIT, it can be stated that the addition of thermally damaged recycled materials leads to oxidative decomposition in a shorter time. In the case of the part made only of recycled material, the oxidation induction time compared to virgin material was reduced by up to 32% (see Figure 8). It follows from the experimental measurements that in studying the detection of recycled material, the method of determination of the induction oxidation time proved to be successful, but this concerned materials containing 30 wt.% of the recycled material and above. This behavior was in agreement with data observed by Schmid et al. [19], Li et al. [20], and Hoyos et al. [21].

### 4.3. Evaluation of the Melt Flow Volume Index (MVR)

The melt flow volume index (MVR) was measured in accordance with EN ISO 1133-1. From the obtained MVR values, a graphical illustration of the dependence of the MVR on the proportion of weight of the degraded recycled material in the virgin material was compiled (see Figure 9).

Figure 9 makes it quite apparent that with the increasing proportion of weight of the recycled material in the mixture, there was a significant increase in the values of the volume index of the melt flow, whereby in 100% recycled material the flowability of the material before exposure increased 2.5 times. The increase in values is logical in this case, because the degradation of the material means cleavage and shortening of the macromolecular chains, which results in an increase in the material’s fluidity. This behavior was in agreement with data observed by Gadgeel et al. [22] and Chen et al. [23].

The experimental measurement of MVR, which is closely related to molecular weight, makes it clear that this method is definitely suitable for the detection of recycled material contained in virgin material, not only in relation to the measured results, but also from a practical point of view, because as a rule device used to determine the melt flow index are available to all major plastics processors, or at least they are able to outsource this measurement as per their customers’ requirements. A decrease in the service life of the parts due to the presence of recycled material is more than obvious from the measured MVR values for ejections exposed to elevated temperature (see Table 5).

### 4.4. Evaluation of Tensile Properties

The courses of tensile properties and the variance of measured values, expressed by the standard deviation, point to the fact that the addition of the thermally damaged recycled material up to 50 wt.% results in a negligible change in tensile properties. The elongation of the material, or its relative elongation at break point, remains unchanged (see Figure 10).

In the case of a part made of only recycled material, the modulus of elasticity decreased compared to the virgin material by only 6% (see Figure 11), the tensile strength (or yield stress) decreased by 4% (see Figure 12), and an increase in the relative elongation at the yield point by 24% occurred (see Figure 13).

It clearly follows from the results for the tensile properties, which were obtained from the samples exposed to conditions of 150 °C/500 h, that with the addition of the recycled material up to 10 wt.%, these properties change, e.g., the yield stress decreases by 60% (see Figure 14), the breaking stress decreases by up to 67% (see Figure 15), and the nominal relative elongation at yield point and at break point (i.e., also the total elongation; see Figure 16 and Figure 17 decreases by almost 80%. The results also confirm the conclusions made in the studies of Liu et al. [24] and Atagur et al. [25].

The results of these properties clearly show that in the short term, the addition of recycled material to the virgin material does not change the tensile properties but significantly affects the life of the part. The results also confirm the conclusions obtained in the studies by Lin et al. [26] and Razak et al. [27].

### 4.5. Evaluation of Bending Properties

Graphical illustrations of the dependence of the flexural strength and modulus of elasticity on the proportion of weight of the recycled material contained in the polymeric material were created. The above-mentioned graphical illustrations of the dependence clearly indicate that the content of the recycled material in the mixture significantly reduces the values of the ejections’ bending characteristics, even at small amounts of up 10 wt.%. In the parts with the degraded recycled material making up 10 wt.%, the flexural strength decreased by almost 12% (see Figure 18) and the modulus of elasticity by as much as 22% (see Figure 19). This behavior was in agreement with data observed by Khademieslam et al. [28] and Hamada et al. [29].

When comparing the values for the tensile modulus (see Figure 11) and flexural modulus (see Figure 19), it is obvious that to detect the recycled material in the polymer, it is more suitable to monitor the modulus of elasticity in bending, where the decrease was detected as soon as the recycled material accounted for 10 wt.% of the polypropylene, while the tensile modulus decreased only in the case of ejections where the recycled material made up 30 wt.%. It is also clear from the results of the bending properties of the ejections exposed to the conditions of 150 °C/500 h that the addition of the recycled material significantly shortens the service life of the part. This fact is evident from the decrease in flexural strength values. In a part made of virgin material and exposed to an elevated temperature of 150 °C for 500 h, the flexural strength decreased by almost 17%; in a part with the recycled material making up 10 wt.%, this was by 82%; and in a part made of purely recycled material, this was down by about 88% (see Figure 20). This behavior was in agreement with data observed by Kim et al. [30] and Wang et al. [31].

### 4.6. Evaluation of Notched Strength Using the Charpy Method

Charpy notched strength values were established on test specimens with a standardized notch of the A type in accordance with the standard CZS EN ISO 179-1. In the case of Charpy notched strength, the addition of thermally damaged recycled material into the virgin material was detected only when the amount was greater than 50 wt.%, whereby a decrease in notched strength of about 11% was recorded (see Figure 21). In the case of a part made only of the recycled material, the Charpy notched strength value decreased by 44%. From the results of the Charpy notched strength of the samples exposed to a temperature of 150 °C for 500 h (simulating the service life of components), it can be stated that the addition of recycled material significantly reduced the service life of the part. In the case of the ejection made of virgin material, the notched Charpy strength remained unchanged after exposure (in terms of scattering of the measured values). In the case of the ejection with the recycled material accounting for 10 wt.%, it decreased by 88% (see Figure 22). These results also confirm the conclusions made in the studies by Khalili et al. [32] and Hernandez-Dias et al. [33].

## 5. Statistical Analysis of the Results

The basic statistical analysis of the general model (1) used to predict the investigated response *y* depending on the change of the investigated independent variables *xi* in nominal (state) and ordinal scales (recycled amount) was performed using the analysis of variance (ANOVA) method. The analysis of variance for the investigated parameter *y* represents a basic statistical analysis of the suitability of the general model used (1).
(1)$$\overset{⌢}{y}=b_{0}.x_{0}+\sum_{j=1}^{N}b_{j}.x_{j}+\sum_{\begin{array}{l} u,j=1\\ u\neq j \end{array}}^{N}b_{uj}.x_{u}.x_{j}$$

On the one hand, the analysis of variance analyzed whether the variability caused by random errors is significantly less than the variability of the measured values explained by the model. The second statistical view of the ANOVA follows from its basic character, where we tested a null statistical hypothesis that none of the effects used in the models (state, amount of recycled material) cause a significant change in the investigated variable (*y*). Within the statistical analysis of the experimental results, we worked with the factor analysis of variance, whereby the influences of the main effects of the independent variables and their mutual interaction were considered. A general table of ANOVA parameters is shown in Table 6.

Within the statistical evaluation of individual examined variables by means of analysis of variance, we came to the conclusion for the monitored dependent variable *acA* that at the selected level of significance *α* = 0.05, there are significant effects for the recycled amount (*p* = 0.000), state (*p* = 0.000), and the interaction of the recycled amount and condition (*p* = 0.000). The influence of the amount of recycled material on the change in the value of the investigated variable *acA* was 26.98%, the influence of the state on the change in the value of the investigated variable as 51.74% and the influence of their mutual interaction was 18.96%. Regarding the mutual comparison of mean values of repeated measurements using Fisher’s individual test of average differences, it can be said that with the exception of the mutual comparison (difference) of 30% and 20% contents of recycled material (*p* = 0.657), the differences in mean values of repeated measurements were −1.495 ± 0.47 (kJ/m^2^), which were statistically insignificant at the significance level of α = 0.05. At the same time, we can state that there was a significant difference in the mean values of repeated measurements of 6.3 ± 0.3 (kJ/m^2^) between the normal state and the state after aging (*p* = 0.000).

As part of the investigation of bending properties, it is possible to conclude that the monitored parameters *Ef* (MPa) and *σfM* (MPa) were significantly influenced by the monitored input variables as the main effects, while their changes at the selected level of significance also significantly affected the changes of the investigated parameters (*α* = 0.05). The influence of the amount of recyclate on the change of the value of the parameter *Ef* was 67.80%, the influence of the state was 11.13%, and finally the influence of the interaction of the input variables on the change of the value of *Ef* was 17.38%. A mutual comparison of the mean values of the individual levels of recycled materials indicated that there were significant differences between the individual values and the mean values of repeated measurements, with differences in the range of 69.0 ± 53.6 (MPa) for contents of 30% and 10% after 629.1 ± 54.9 (MPa) and with additional differences between 50% and 0%. From the point of view of a mutual comparison of the state, there was a significant difference at the selected level of significance *α* = 0.05 between the mean value of repeated measurements between the normal state and the state after aging at the level of—160.4 ± 31.4 (MPa). Additionally, the influence of the amount of recycled material on the change in the value of the parameter *σfM* was 19.17%, the influence of the state was 72.30%, and the influence of the mutual interaction was 7.84%. By comparing the differences of the mean values of repeated measurements at individual levels of the amount of recycled material, these differences were significant at the level of significance *α* = 0.05, with the exception of mutual differences of 30% to 10% (*p* = 0.275), 30% to 20% (*p* = 0.199), 50% to 20% (*p* = 0.313), 100% to 20% (*p* = 0.250), and 100% to 50% (*p* = 0.885). Additionally, the difference in the mean values of repeated measurements using Fisher test between the normal state and the state after aging showed that this difference was significant (*p* = 0.000) at the selected level of significance *α = 0.05* and reached a value of 27.327 ± 0.892 (MPa).

The analysis of tensile properties expressed by parameters *σm* (MPa), *εm* (%), *εB* (%), and *Et* (MPa) by ANOVA pointed to the fact that independent variables (amount of recyclate, state) and their mutual interaction significantly influence the change of values of tensile properties examined using the parameters *σm* (MPa), *εm* (%), *εB* (%), and *Et* (MPa) at the selected significance level of *α = 0.05*. The effect of the amount of recycled material on the change in *σm* (MPa) was 7.32%, on the value of *εm* (%) was 1.06%, on the change in *εB* (%) was 1.72%, and on the change in *Et* (MPa) was 27.52%. The influence of the state on the change of the value of the investigated variable *σm* (MPa) was 84.84%, on the value of *εm* (%) was 95.42%, on the change of the value of *εB* (%) was 87.24%, and on the change of the value of *Et* (MPa) was 42.30%. The influence of the mutual interaction of the amount of recycled material and the state on the change of the value of the investigated variable *σm* (MPa) was 1.22%, on the value of *εm* (%) was 2.98%, on the change of the value of *εB* (%) was 1.89%, and on the change of the value *Et* (MPa) was 24.71%.

## 6. Conclusions

This paper dealt with the possibility of detecting a recycled material contained in a polymer, namely a polypropylene or ethylene–propylene copolymer. The polypropylene SABIC^®^ PP CX02-82 was selected for experimental measurements. For experimental research purposes, the recycled material was prepared from the above material, which was intentionally degraded through increased temperature of the melt during injection (verified via a 2.5-fold increase in the melt flow index compared to the virgin material). The recycled material was added to the virgin material at a specified weight ratio, while the multi-purpose test specimens were subsequently injected from the raw material prepared under conditions corresponding to the regulations in EN ISO 1873-2 and EN ISO 294-1.

Techniques commonly used in experimental industrial practice were used in the study, which made it possible to establish the flow properties of the material melts, their oxidative stability, and also their thermal and mechanical properties.

Based on the methods used, it can be clearly stated that the most suitable technique for detecting thermally degraded recycled material in polypropylene is the method of establishing the melt flow index according to EN ISO 1133. With the recycled content making up 10 wt.%, the value of the melt flow volume index (MVR) increased by approximately 14%, while at 20 wt.% of the recycled material, it increased by 24% (due to a change in the molecular structure of the material or due to a decrease in its molecular weight).

The bending test (EN ISO 178) seems to be the most suitable method for detecting recycled material by measuring the material’s mechanical properties. Similarly to the MVR method, it was possible to unambiguously detect the presence of even a small amount of recycled material in the whole part by measuring the material’s bending properties. At 10 wt.%, the heat-damaged recycled material accounted for the polypropylene total wt., a decrease in flexural strength of 12%, while a flexural modulus of 22% was recorded in the studied samples (at 20 wt.% of the recycled material, the flexural strength decreased by 14% and the modulus of elasticity by 26%). None of the other techniques used (for the evaluation of mechanical properties) allowed unambiguous detection of the recycled material if it accounted for less than 20 wt.%. Changes in the tensile properties of the studied samples were negligible with regard to the variance of the measured values, while a clear change in the Charpy notched strength was recorded only when the proportion of the recycled material in the virgin material was over 50 wt.%.

The differential scanning calorimetry technique was used to measure the thermal properties of the material, which are a reflection of its structure, and also to determine the induction oxidation time at a constant temperature in an oxygen atmosphere. Based on the acquired knowledge, it can be stated that for monitoring the presence of a recycled or degraded material, a more suitable method is that of establishing the OIT than the classical DSC measurement, which results in transition temperatures and enthalpy values of phase transformations (melting and crystallization). By measuring the OIT value, the unambiguous presence of the recycled material was detected only when its content was 30 wt.%. In the classical DSC method, a decrease in the value of the enthalpy of melting was observed, indicating a lower degree of crystallinity of the ejection, which was observable only when the recycled content reached 50 wt.%.

All the techniques used clearly showed that the addition of degraded recycled material reduced the service life of the part, even at as low a content as that of 10 wt.%. This fact was simulated by thermal exposure of the samples to a temperature of 150 °C for 500 h, where in the case of a part made of virgin material, the flexural strength decreased by almost 17%, in the case of a part with 10 wt.% of the recycled material it was reduced by 82%, and in a part made of purely recycled material it was reduced by approximately 88%. It is clear from the results that in the short term, there may be no change in the useful properties of the parts, however in the long term the presence of degraded recycled material will have adverse consequences on their lifespan.

Recommendations for further research include an experimental study on detecting recycled materials in polypropylene, involving degraded recycled materials. It would, therefore, be appropriate to use the recommended techniques for detecting recycled materials (MVR, flexural modulus measurement, and the OIT determination) and to apply them to polypropylene filled with common process waste (thermally undamaged) or to reinforced polypropylene containing a commonly used talc-based mineral filler.

## Figures and Tables

**Figure 1 materials-14-00552-f001:**
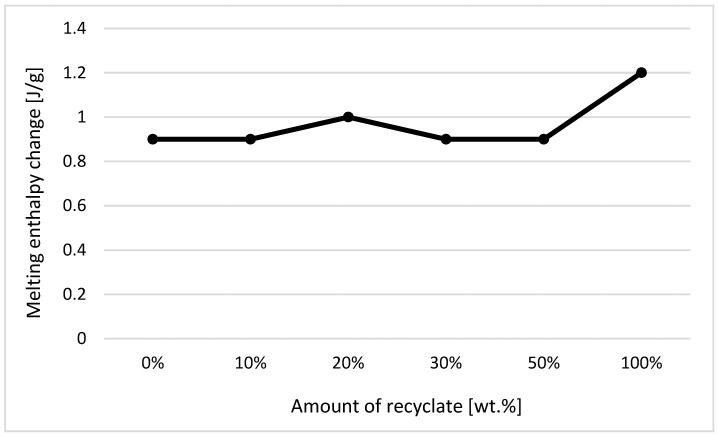
Dependence of the change in melting enthalpy on the proportion of weight of the recycled material before exposure of the samples to elevated temperature—component 1.

**Figure 2 materials-14-00552-f002:**
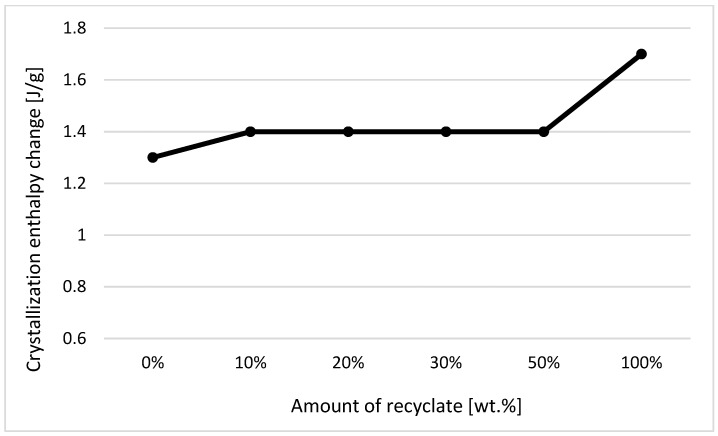
Dependence of the change in enthalpy of crystallization on the proportion of weight of the recycled material before exposure of samples to elevated temperature—component 1.

**Figure 3 materials-14-00552-f003:**
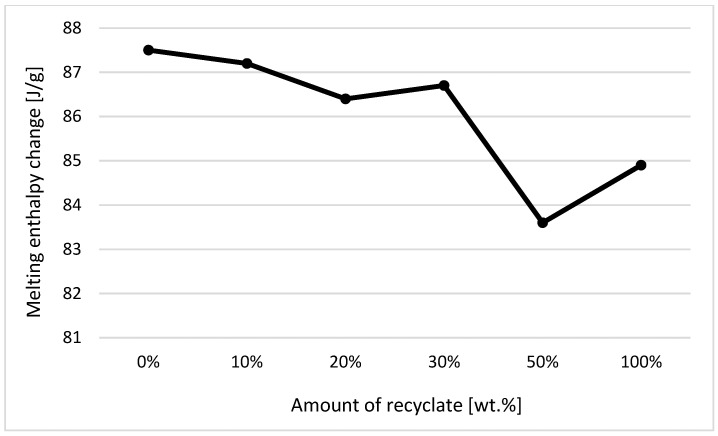
Dependence of the change in melting enthalpy on the proportion of weight of the recycled material before exposure of the samples to elevated temperature—component 2.

**Figure 4 materials-14-00552-f004:**
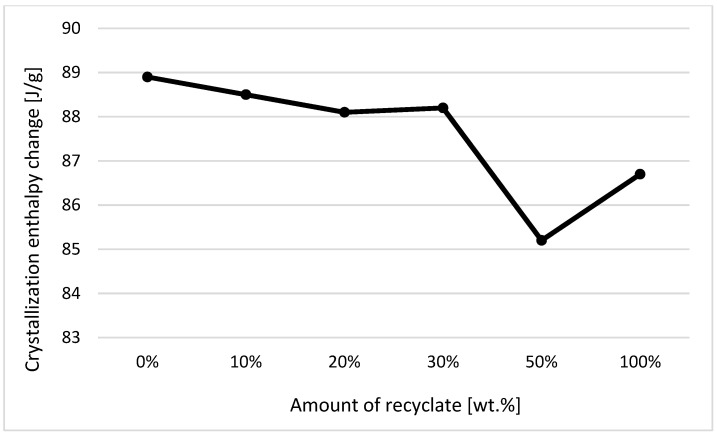
Dependence of the change in enthalpy of crystallization on the proportion of weight of the recycled material before exposure of samples to elevated temperature—component 2.

**Figure 5 materials-14-00552-f005:**
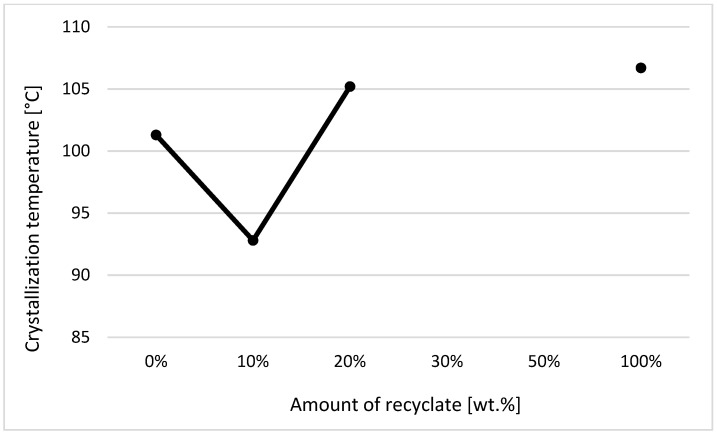
Dependence of the crystallization temperature on the proportion of weigh of the recycled material after exposure of samples to elevated temperature—component 1.

**Figure 6 materials-14-00552-f006:**
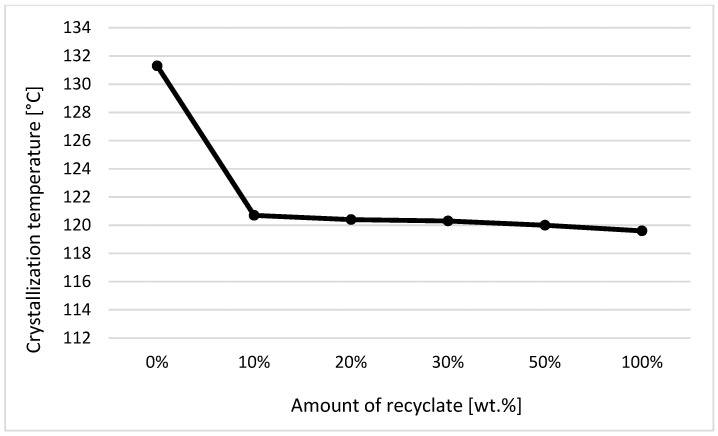
Dependence of the crystallization temperature on the proportion of weight of the recycled material after exposure of samples to elevated temperature—component 2.

**Figure 7 materials-14-00552-f007:**
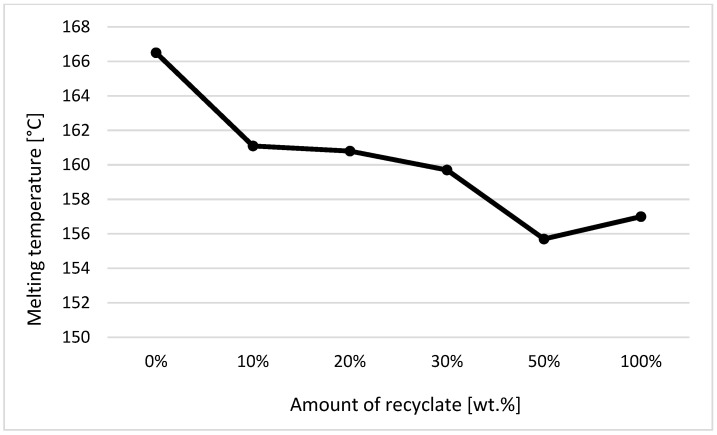
Dependence of the melting temperature on the proportion of weight of the recycled material after exposure of the samples to elevated temperature—component 2.

**Figure 8 materials-14-00552-f008:**
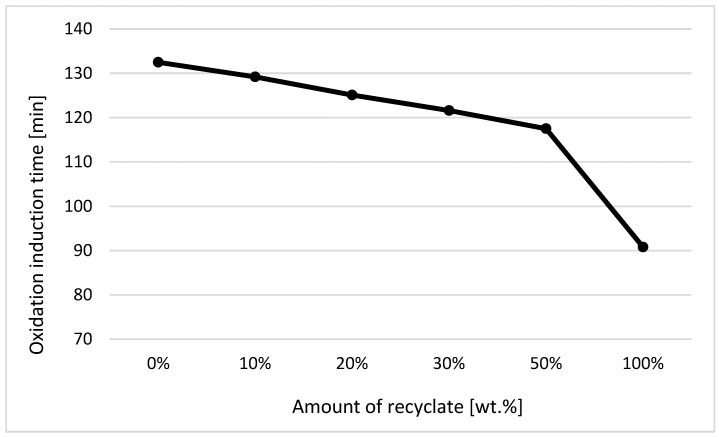
Dependence of the oxidation induction time (OIT) on the proportion of weight of the recycled material before exposure of the samples to elevated temperature.

**Figure 9 materials-14-00552-f009:**
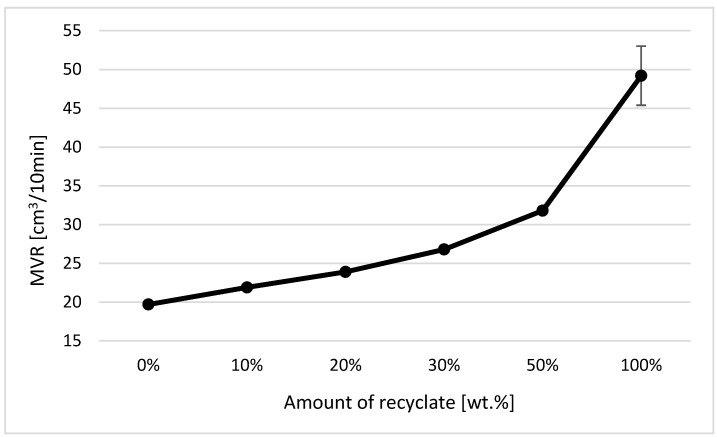
Dependence of the melt flow volume index on the proportion of weight of the recycled material prior to exposure.

**Figure 10 materials-14-00552-f010:**
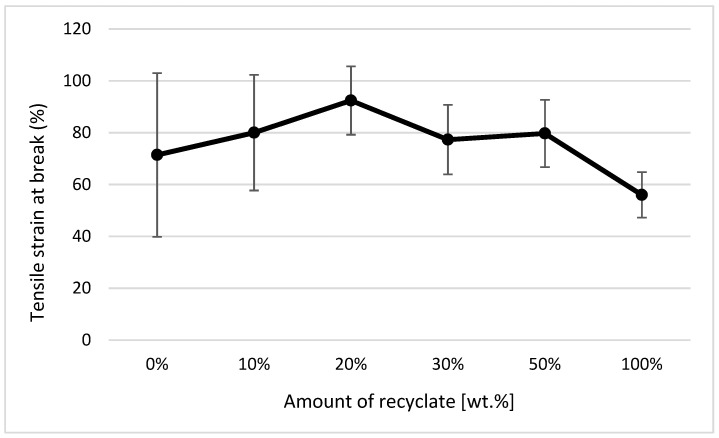
Dependence of the elongation at break point on the proportion of weight of the recycled material before exposing the samples to elevated temperature.

**Figure 11 materials-14-00552-f011:**
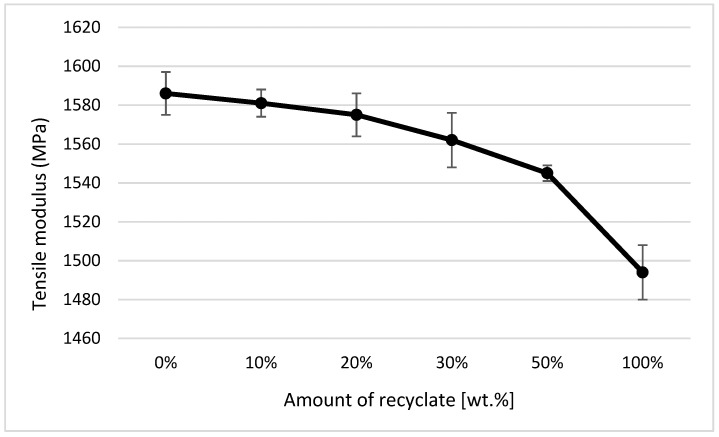
Dependence of the modulus of elasticity on the proportion of weight of the recycled material before exposure of samples to elevated temperature.

**Figure 12 materials-14-00552-f012:**
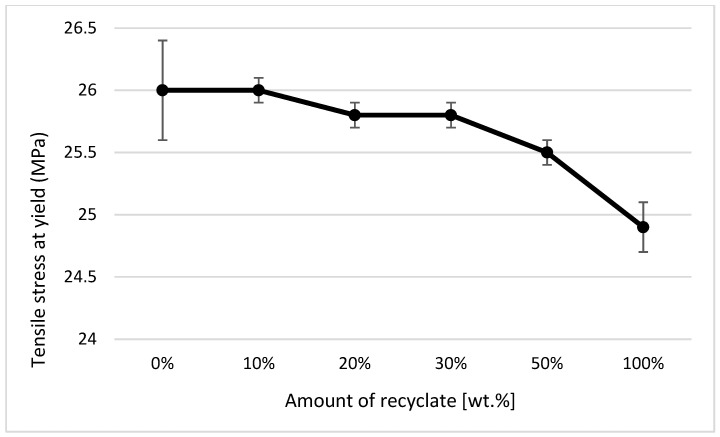
Dependence of yield stress (strength limit) on the proportion of weight of the recycled material before exposure of samples to elevated temperature.

**Figure 13 materials-14-00552-f013:**
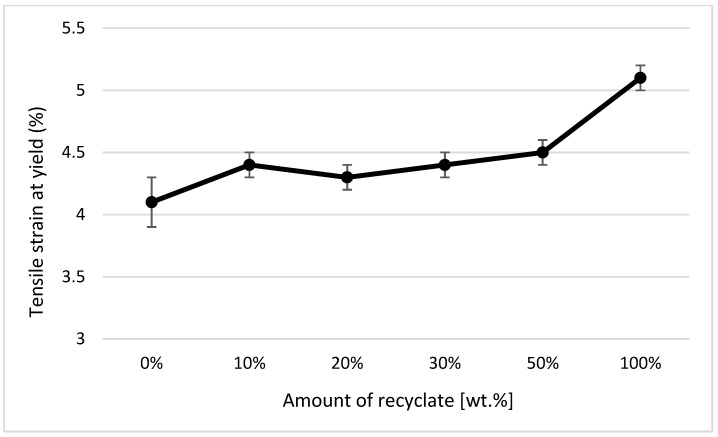
Dependence of the relative elongation at yield strength on the proportion of weight of the recycled material before exposing the samples to elevated temperature.

**Figure 14 materials-14-00552-f014:**
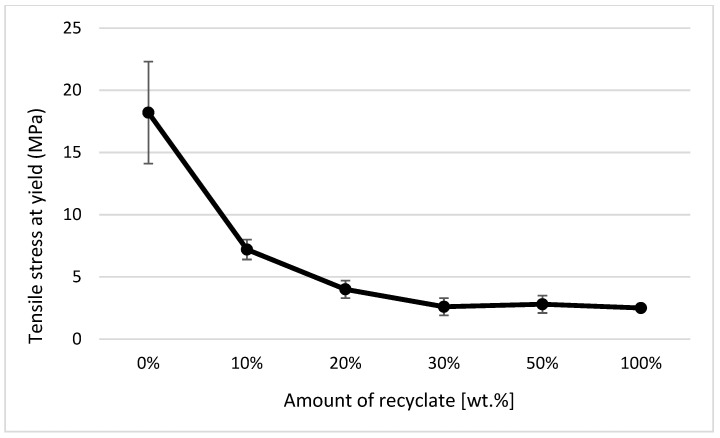
Dependence of stress at yield strength (strength limit) on the proportion of weight of the recycled material after exposure of samples to elevated temperature.

**Figure 15 materials-14-00552-f015:**
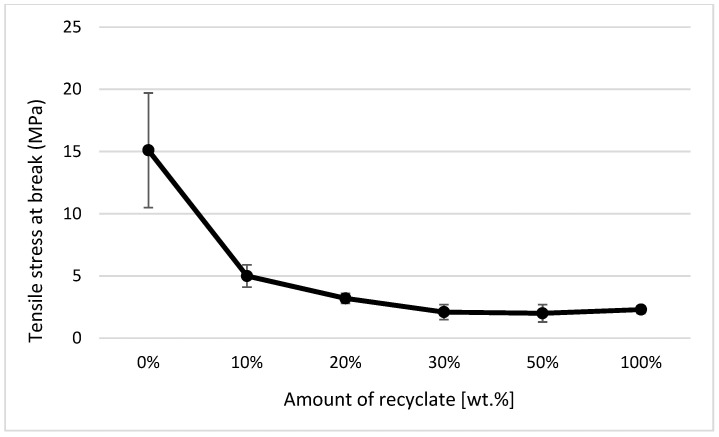
Dependence of stress at the break point on the proportion of weight of the recycled material after exposure of samples to elevated temperature.

**Figure 16 materials-14-00552-f016:**
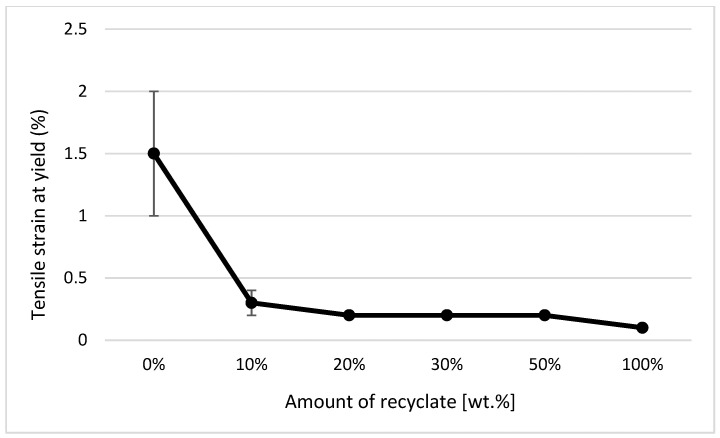
Dependence of stress at the break point on the proportion of weight of the recycled material after exposure of samples to elevated temperature.

**Figure 17 materials-14-00552-f017:**
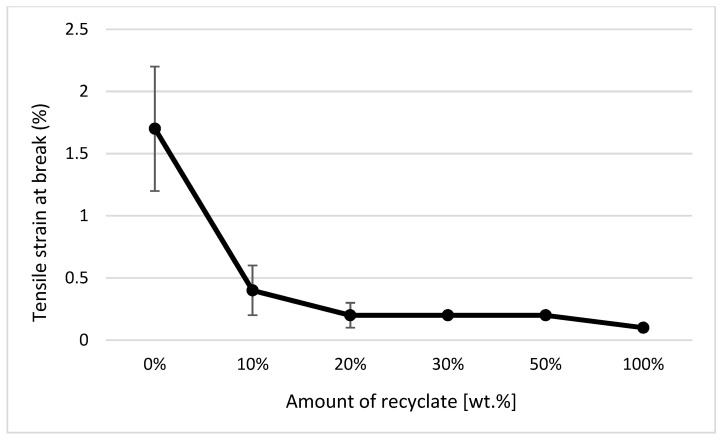
Dependence of the elongation at break point on the proportion of weight of the recycled material after exposure of the samples to elevated temperature.

**Figure 18 materials-14-00552-f018:**
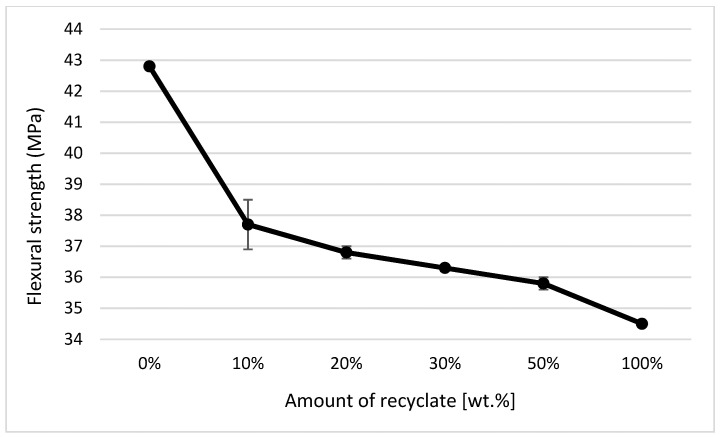
Dependence of the flexural strength on the amount of the recycled material before exposing the samples to elevated temperature.

**Figure 19 materials-14-00552-f019:**
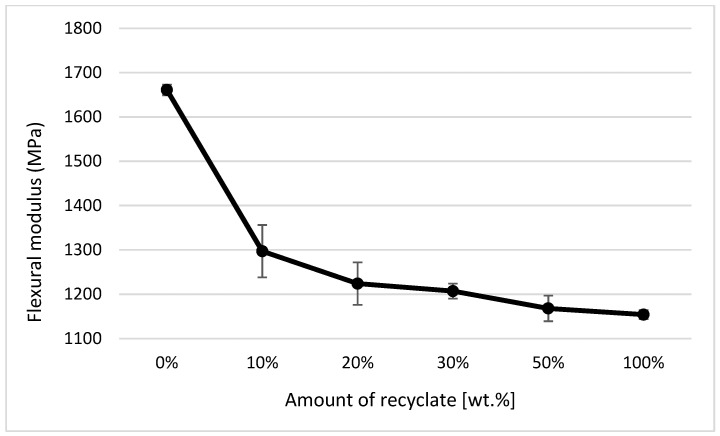
Dependence of the flexural modulus on the amount of recycled material before exposing the samples to elevated temperature.

**Figure 20 materials-14-00552-f020:**
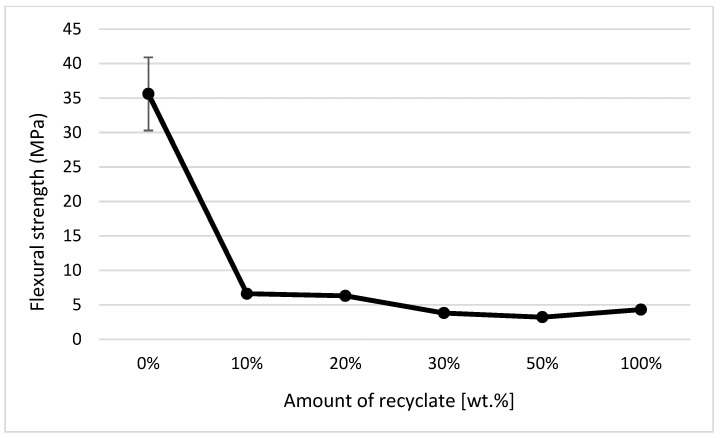
Dependence of the flexural strength on the amount of the recycled material after exposure of samples to elevated temperature.

**Figure 21 materials-14-00552-f021:**
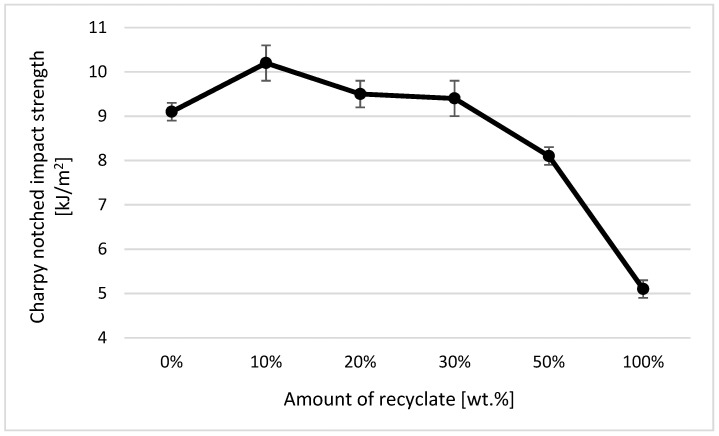
Dependence of the notched strength on the amount of recycled material before exposing samples to elevated temperature.

**Figure 22 materials-14-00552-f022:**
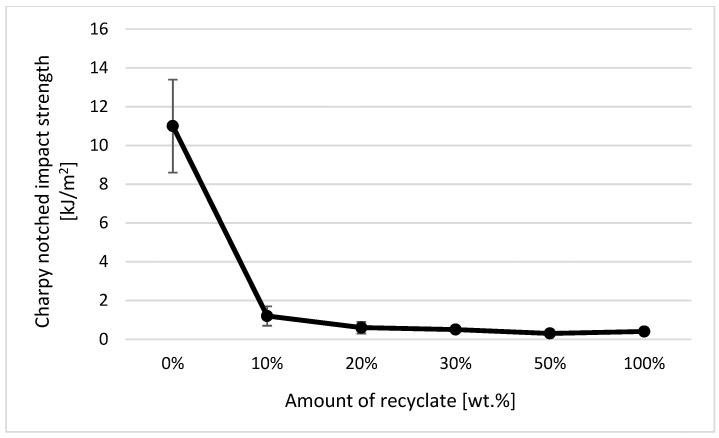
Dependence of the notched strength on the amount of the recycled material after exposure of the samples to elevated temperature.

**Table 1 materials-14-00552-t001:** Typical properties of SABIC^®^ PP CX02-82 polypropylene copolymer.

Properties	Value	Unit
Density	0.905	g/cm^3^
Melt mass flow rate (230 °C; 2.16 kg)	15	g/10 min
Melt volume flow rate (230 °C; 2.16 kg)	19	cm^3^/10 min
Tensile stress at yield	27	MPa
Tensile strain at yield	4	%
Tensile modulus	1550	MPa
Charpy notched impact strength	13	kJ/m^2^
Shore hardness	65	

**Table 2 materials-14-00552-t002:** Changes in the volume melt flow index (MVR) values depending on melt temperature.

Melt Temperature [°C]	MVR [cm^3^/10 min]	Increase of MVR
200	19.7 ± 0.1	-
280	23.2 ± 0.1	1.2 ×
300	32.3 ± 0.2	1.6 ×
310	50.4 ± 0.4	2.6 ×
330	75.7 ± 0.6	3.8 ×

**Table 3 materials-14-00552-t003:** Technological conditions for the injection of samples into the recycled materials.

Properties	Value	Unit
Melt temperature	310	°C
Temperature of the mold tempering medium	40	°C
Cycle time	60	s
Holding time	40	s
Holding pressure	40	MPa
Injection capacity	40	cm^3^
Pressure switch point	16	cm^3^
Injection speed	30	cm^3^/s

**Table 4 materials-14-00552-t004:** Technological conditions for injecting test specimens.

Properties	Value	Unit
Melt temperature	200	°C
Temperature of the mold tempering medium	40	°C
Cycle time	60	s
Holding time	40	s
Holding pressure	40	MPa
Injection capacity	40	cm^3^
Pressure switch point	16	cm^3^
Injection speed	30	cm^3^/s

**Table 5 materials-14-00552-t005:** Mean values for the melt flow volume index after exposure of the samples to elevated temperature.

Batch (wt.% of Recyclate)	Before Exposure MVR (cm^3^/10 min)	After Exposure MVR (cm^3^/10 min)
0	19.7 ± 0.1	64 ± 6
10	21.9 ± 0.1	595 ± 117
20	23.9 ± 0.1	1419 ± 72
30	26.8 ± 0.2	immeasurable
50	31.8 ± 0.1	immeasurable
100	49.2 ± 3.8	immeasurable

**Table 6 materials-14-00552-t006:** General table of ANOVA parameters.

Source	DF	Sum of Squares	Mean Square	F Ratio	Prob > F
Model	*DF*_Model_ = *a* − 1	*S* _Model_	*MS*_Model_ = *S*_Model_/*DF*_Model_	*F* = *MS*_Model_/*MS*_Error_	*p_M_*
Error	*DF*_Error_ = *N* − *a*	*S* _Error_	*MS*_Error_ = *S*_Error_/*DF*_Error_		
C. Total	*DF*_C_._Total_ = *N* − 1	*S*_C_._Total_	*MS*_C_._Total_ = *S*_C_._Total_/*DF*_C_._Total_		

## Data Availability

Data sharing is not applicable to this article.

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
