# Peer review of "Application of Physical Methods for the Detection of a Thermally Degraded Recycled Material in Plastic Parts Made of Polypropylene Copolymer"

_materials, 2021, doi:10.3390/ma14030552_

Round 1
Reviewer 1 Report
This manuscript considered the comparative study to detect thermally degraded recycled material in PP. Below are some comments for the revision.
- In the Introduction part, the objective and scope of this study should be included. The present version only introduced the literature review in the Introduction.
- As a rheologist, this reviewer recommend the comparison of rheological properties (like shear viscosity, or G' and G'') for polymers with recycled materials.
- Many similar figures in the DSC and tensile results manuscript can be efficiently reduced.
- The following results should be further discussed. (1) In Figure 10, tensile strain at break was the maximum for 20% sample. (2) In Figure 21, Charpy notched impact strength was the maximum in 10% sample.
Author Response
Dear reviewer,
thank you very much for your comments on the article. In this annex I attach the changes made to the article. I respond to your comments below:
- In the Introduction part, the objective and scope of this study should be included. The present version only introduced the literature review in the Introduction.
A brief description of the aim and content of the article was added at the beginning of the first chapter - Introduction.
- As a rheologist, this reviewer recommend the comparison of rheological properties (like shear viscosity, or G' and G'') for polymers with recycled materials.
The aim of the paper was to assess the possibilities of using "basic" laboratory methods, which would allow in technical practice to prove the non-conformity of plastic parts due to the presence of thermally damaged recycled material in virgin material. We agree with the reviewer that the results of rheological properties characterized by shear viscosity or storage modulus (G´) and loss modulus (G´´) would be of scientific interest. Rotary rheometers and dynamic-mechanical analysis are not among the standard equipment of industrial laboratories, therefore a capillary rheometer was used for the experimental study, ie. methods for determining the melt flow index, which is usually used in the input or output quality control of the polymer. The reviewer's initiative will be incorporated into a separate article dealing with the rheological evaluation of the polymer depending on the addition of recyclate, where the rheological properties of the polymer are assessed before and after its thermal and atmospheric aging. This data set will be very extensive and would disproportionately prolong the present article, so the results of the study will be published separately.
- Many similar figures in the DSC and tensile results manuscript can be efficiently reduced.
We apologize, but we think that all the images mentioned in the article should be kept to better explain the results. We would like to keep these results in their current form. Thank you for your understanding.
- The following results should be further discussed. (1) In Figure 10, tensile strain at break was the maximum for 20% sample. (2) In Figure 21, Charpy notched impact strength was the maximum in 10% sample.
Due to the variance of the measured values of tensile strain at break (see Fig. 10), the differences in tensile strain at break (total elongation) are statically insignificant and it cannot be clearly stated that the maximum elongation was recorded for the 20% recycled material. The change in Charpy notch toughness for the 10% recycled material corresponds to the change in crystallization temperature (see Fig. 5), where the material crystallizes at a lower temperature, which probably resulted in a structural (morphological) change in the material causing an increase in its toughness.

Reviewer 2 Report
Please see the attached file.

Author Response
Reviewer 2
Dear reviewer,
thank you very much for your comments on the article. In this annex I attach the changes made to the article. I respond to your comments below:
Line 14 – corrected
Line 20 – corrected
- MVR – Melt Volume Flow Rate
Line 126 – corrected
- plastic components
Line 127-129
Q: Any reference to back-up this?
A: No reference.
Line 129-133 – corrected
Line 137-138
Q: Please define.
A: This can happen if the injection molding machine operator enters incorrect technological parameters by mistake.
Line 144
Q: Why would anyone process PP at such elevated temperatures? The degradation of PP can be measured using multiple features.
A: This temperature was used only purposely for the preparation of thermally degraded recycled material.
Line 157
Q: OK, so you are investigating the properties of PP exposed to extreme temperatures?
A: No. This temperature was used only purposely for the preparation of thermally degraded recycled material as part of the experimental research.
Line 174 – corrected
- 23/50 (Temperature/Relative humidity)
Line 180-187 – corrected
Line 188 – corrected
Q: Is this about methods or results?
A: Results.
Q: I think, that the deviations for the measured results are required.
A: No. Repeated measurements were not performed.
Line 279-281 – corrected
Line 320
Q: Are the finding statistically significant? (Not?)
A: Due to the variance of the measured values of Tensile Strain at Break, which is expressed by the standard deviation, the influence of the amount of thermal damage of the recycled material on the measured physical quantity is statically insignificant. The results indicate that the identification of the recycled material in the polymeric material based on the Tensile Strain at Break determination has not been clearly demonstrated. This finding is important for technical practice in selecting a suitable method (technique) that will detect the addition of recycled material in the virgin polymer and its adverse effect on the physical and structural properties of the part. For this reason, the results are part of an expert article, and the text in Chapter 5, which summarizes the conclusions reached, has also been modified in this sense.
Line 433
Q: This is very simple, *IF* you know you are dealing with PP.
A: Yes. You are right.
Line 439
Q: a good find
A: Thank you.
Line 470
Q: This is a significant part of the study, as you have just partially opened the 'Pandoras Box' that is relevant to the polymer recycling.
A: Thank you.

Round 2
Reviewer 1 Report
Comments raised by this reviewer were fully considered in the revised version.
Author Response
Comments raised by this reviewer were fully considered in the revised version.
Reviewer 2 Report
Unfortunately I do not find the statistical analysis of the results as repetitions were not conducted. Hence, I do not find the paper scientifically sound.
Author Response
Reviewer 2_Round 2
Dear reviewer,
thank you very much for your comments on the article. The required statistical analysis has been added to the newly added chapter of the article entitled "Statistical analysis of the results". The text inserted into the article is below:
- Statistical analysis of the results
The basic statistical analysis of the general model (1) used to predict the investigated response y depending on the change of the investigated independent variables xi in nominal (state) and ordinal scale (recycled amount) is performed using Analysis of variance. The analysis of variance for the investigated parameter y represents a basic statistical analysis of the suitability of the general model used (1).
(1)
On the one hand, analysis of variance analyzes whether the variability caused by random errors is significantly less than the variability of the measured values explained by the model. The second statistical view of the ANOVA follows from its basic character, where we test a null statistical hypothesis that none of the effects used in the models (state, amount of recycled material) affect a significant change in the investigated variable (y). Within the statistical analysis of the experimental results, we worked with the factor analysis of variance, where the influence of the main effects of the independent variables and their mutual interaction was considered. The general table ANOVA is shown in Table 6.
Table 6. General table ANOVA.
|
Source |
DF |
Sum of Squares |
Mean Square |
F Ratio |
Prob > F |
|
Model |
DFModel=a-1 |
SModel |
MSModel= SModel/DFModel |
F= MSModel/MSError |
pM |
|
Error |
DFError=N-a |
SError |
MSError= SError/DFError |
||
|
C. Total |
DFC.Total=N-1 |
SC.Total |
MSC.Total= SC.Total/DFC.Total |
Within the statistical evaluation of individual examined variables by means of analysis on variance, we came to the conclusion for the monitored dependent variable acA that at the selected level of significance α = 0.05 there is a significant effect of recycled amount (p = 0.000), state (p = 0.000) as well as the interaction of recycled amount and condition (p = 0.000). The influence of the amount of recycled material on the change in the value of the investigated variable acA represents 26.98%, the influence of the state on the change in the value of the investigated variable represents 51.74% and the influence of their mutual interaction is 18.96%. From the point of view of mutual comparison of mean values of repeated measurements using Fisher's individual test of average differences, it can be said that with the exception of mutual comparison (difference) of 30% and 20% of recycled amount (p = 0.657) the difference of mean values of repeated measurements is -1.495 ± 0.47 [kJ/m2] statistically insignificant at the significance level α = 0.05. At the same time, we can state that there is a significant difference in the mean values of repeated measurements of 6.3 ± 0.3 [kJ/m2] between the normal state and the state after aging (p = 0.000).
As part of the investigation of bending properties, it is possible to conclude that the monitored parameters Ef [MPa] and σfM [MPa] are significantly influenced by the monitored input variables as the main effects and their change at the selected level of significance also significantly affects the change of investigated parameters α = 0,05. The influence of the amount of recyclate on the change of the value of the parameter Ef represents 67.80%, the influence of the state represents 11.13% and finally the influence of the interaction of the input variables on the change of the value of Ef represents 17.38%. A mutual comparison of the mean values of the individual levels of recycled amount indicates the fact that there is a significant difference between the individual values of the mean values of repeated measurements, these differences ranging from -69.0 ± 53.6 [MPa] with a difference between 30% and 10% after - 629.1 ± 54.9 [MPa] with a difference between 50% and 0%. From the point of view of mutual comparison of the state, there is a significant difference at the selected level of significance α = 0.05 between the mean value of repeated measurements between the normal state and the state after aging at the level of -160.4 ± 31.4 [MPa]. Also, the influence of the amount of recycled material on the change in the value of the parameter σfM represents the value of 19.17%, the influence of the state represents 72.30% and the influence of the mutual interaction represents the value of 7.84%. By comparing the differences of the mean values of repeated measurements of individual levels of the amount of recycled material, this difference is significant at the level of significance α = 0.05, with the exception of mutual differences of 30% - 10% (p = 0.275), 30% - 20% (p = 0.199), 50% - 20% (p = 0.313), 100% - 20% (p = 0.250) and 100% - 50% (p = 0.885). Also, the difference in the mean values of repeated measurements using the Fisher test between the normal state and the state after aging shows that this difference is significant (p = 0.000) at the selected level of significance α = 0.05 and reaches the value 27.327 ± 0.892 [MPa].
Analysis of tensile properties expressed by parameters σm [MPa], εm [%], εB [%] and Et [MPa] by ANOVA points to the fact that independent variables (amount of recyclate, state) and their mutual interaction significantly influence the change of values of tensile properties examined using the parameters σm [MPa], εm [%], εB [%] and Et [MPa] at the selected significance level α = 0.05. The effect of the amount of recycled material on the change in σm [MPa] is 7.32%, on the value of εm [%] is 1.06%, on the change in εB [%] is 1.72% and on the change in Et [MPa] is 27.52%. The influence of the state on the change of the value of the investigated variable σm [MPa] represents 84.84%, on the value of εm [%] is 95.42%, on the change of the value of εB [%] is 87.24% and on the change of the value of Et [MPa] is 42.30%. The influence of the mutual interaction of the amount of recycled material and the state on the change of the value of the investigated variable σm [MPa] is 1.22%, on the value of εm [%] is 2.98%, on the change of the value of εB [%] is 1.89% and on the change of the value Et [MPa] is 24.71%.
Reviewer 2_Round 2
Dear reviewer,
thank you very much for your comments on the article. The required statistical analysis has been added to the newly added chapter of the article entitled "Statistical analysis of the results". The text inserted into the article is below:
- Statistical analysis of the results
The basic statistical analysis of the general model (1) used to predict the investigated response y depending on the change of the investigated independent variables xi in nominal (state) and ordinal scale (recycled amount) is performed using Analysis of variance. The analysis of variance for the investigated parameter y represents a basic statistical analysis of the suitability of the general model used (1).
(1)
On the one hand, analysis of variance analyzes whether the variability caused by random errors is significantly less than the variability of the measured values explained by the model. The second statistical view of the ANOVA follows from its basic character, where we test a null statistical hypothesis that none of the effects used in the models (state, amount of recycled material) affect a significant change in the investigated variable (y). Within the statistical analysis of the experimental results, we worked with the factor analysis of variance, where the influence of the main effects of the independent variables and their mutual interaction was considered. The general table ANOVA is shown in Table 6.
Table 6. General table ANOVA.
|
Source |
DF |
Sum of Squares |
Mean Square |
F Ratio |
Prob > F |
|
Model |
DFModel=a-1 |
SModel |
MSModel= SModel/DFModel |
F= MSModel/MSError |
pM |
|
Error |
DFError=N-a |
SError |
MSError= SError/DFError |
||
|
C. Total |
DFC.Total=N-1 |
SC.Total |
MSC.Total= SC.Total/DFC.Total |
Within the statistical evaluation of individual examined variables by means of analysis on variance, we came to the conclusion for the monitored dependent variable acA that at the selected level of significance α = 0.05 there is a significant effect of recycled amount (p = 0.000), state (p = 0.000) as well as the interaction of recycled amount and condition (p = 0.000). The influence of the amount of recycled material on the change in the value of the investigated variable acA represents 26.98%, the influence of the state on the change in the value of the investigated variable represents 51.74% and the influence of their mutual interaction is 18.96%. From the point of view of mutual comparison of mean values of repeated measurements using Fisher's individual test of average differences, it can be said that with the exception of mutual comparison (difference) of 30% and 20% of recycled amount (p = 0.657) the difference of mean values of repeated measurements is -1.495 ± 0.47 [kJ/m2] statistically insignificant at the significance level α = 0.05. At the same time, we can state that there is a significant difference in the mean values of repeated measurements of 6.3 ± 0.3 [kJ/m2] between the normal state and the state after aging (p = 0.000).
As part of the investigation of bending properties, it is possible to conclude that the monitored parameters Ef [MPa] and σfM [MPa] are significantly influenced by the monitored input variables as the main effects and their change at the selected level of significance also significantly affects the change of investigated parameters α = 0,05. The influence of the amount of recyclate on the change of the value of the parameter Ef represents 67.80%, the influence of the state represents 11.13% and finally the influence of the interaction of the input variables on the change of the value of Ef represents 17.38%. A mutual comparison of the mean values of the individual levels of recycled amount indicates the fact that there is a significant difference between the individual values of the mean values of repeated measurements, these differences ranging from -69.0 ± 53.6 [MPa] with a difference between 30% and 10% after - 629.1 ± 54.9 [MPa] with a difference between 50% and 0%. From the point of view of mutual comparison of the state, there is a significant difference at the selected level of significance α = 0.05 between the mean value of repeated measurements between the normal state and the state after aging at the level of -160.4 ± 31.4 [MPa]. Also, the influence of the amount of recycled material on the change in the value of the parameter σfM represents the value of 19.17%, the influence of the state represents 72.30% and the influence of the mutual interaction represents the value of 7.84%. By comparing the differences of the mean values of repeated measurements of individual levels of the amount of recycled material, this difference is significant at the level of significance α = 0.05, with the exception of mutual differences of 30% - 10% (p = 0.275), 30% - 20% (p = 0.199), 50% - 20% (p = 0.313), 100% - 20% (p = 0.250) and 100% - 50% (p = 0.885). Also, the difference in the mean values of repeated measurements using the Fisher test between the normal state and the state after aging shows that this difference is significant (p = 0.000) at the selected level of significance α = 0.05 and reaches the value 27.327 ± 0.892 [MPa].
Analysis of tensile properties expressed by parameters σm [MPa], εm [%], εB [%] and Et [MPa] by ANOVA points to the fact that independent variables (amount of recyclate, state) and their mutual interaction significantly influence the change of values of tensile properties examined using the parameters σm [MPa], εm [%], εB [%] and Et [MPa] at the selected significance level α = 0.05. The effect of the amount of recycled material on the change in σm [MPa] is 7.32%, on the value of εm [%] is 1.06%, on the change in εB [%] is 1.72% and on the change in Et [MPa] is 27.52%. The influence of the state on the change of the value of the investigated variable σm [MPa] represents 84.84%, on the value of εm [%] is 95.42%, on the change of the value of εB [%] is 87.24% and on the change of the value of Et [MPa] is 42.30%. The influence of the mutual interaction of the amount of recycled material and the state on the change of the value of the investigated variable σm [MPa] is 1.22%, on the value of εm [%] is 2.98%, on the change of the value of εB [%] is 1.89% and on the change of the value Et [MPa] is 24.71%.

Round 3
Reviewer 2 Report
You have done a lot of work.